# Metaphorical framing of the COVID-19 pandemic in Pakistan: A corpus driven critical analysis of war metaphors in news media

**Arooj Rana**[☯], **Tahir Ayoub**[☯], **Shazia Akbar Ghilzai**[ID]*[☯], **Wasima Shehzad**[☯]

Department of English, Quaid-e-Azam University, Islamabad, Pakistan

☯ These authors contributed equally to this work.
* sghilzai@qau.edu.pk

**Data Availability Statement:** This data can be found in the editorials of Dawn, The Express Tribune, and The News from 15th March to 15th July 2020. The data may not be available in a public

## Abstract

Metaphors are an essential part of how humans process and understand the world. Cognitive linguistics does not view metaphors as merely linguistic or rhetorical devices; rather, they are conceptual in nature and are central to the thought process. Therefore, the present research investigates the metaphorical depiction of the Covid-19 health emergency through the conceptual metaphor of WAR in three renowned Pakistani English Newspapers i.e. Dawn, The Express Tribune, and The News. Critical Metaphor Analysis (CMA) is specifically selected to uncover the covert and possibly unconscious intentions of language users in Newspaper discourse. Fifty (50) editorials on the subject of Covid-19 are specifically chosen and their language is meticulously observed by making a specialized Corpus PakNCovid-19. The size of the corpus is 17621 words. Moreover, Monoconc Corpus Tool is utilized to analyze the metaphorical depiction of Covid-19 as a WAR in Pakistani Newspaper discourse. The study highlights the explicit deployment of military concepts like BATTLE, ENEMY, WAR, SOLDIERS, FIGHT, and VICTORY to create the conception of WAR and to create SELF Vs OTHER distinctions between the Pakistani people and the medical illness of Covid-19. The inquiry demonstrates that to create a sense of urgency and to mobilize masses against the deadly virus, the metaphors of War have been used deliberately. The military concepts have been purposely employed to present Covid-19 as an 'alien', 'outsider', as well as an 'enemy' entity.

## Introduction

Coronavirus disease 2019 (Covid-19) was a globally dispersed respiratory illness caused by a novel virus known as Severe Acute Respiratory Syndrome Coronavirus 2 (SARS-2). The virus was first out broke in December 2019 in Wuhan, Hubei Province, China, and disseminated around the globe [1]. Owing to the rampant widespread of disease, the World Health Organization (WHO) declared it a Public Health Emergency of International Concern on 30th January 2020 [2]. The deadliest virus, due to its high transmission capability as well as high mobility and mortality, had been spread to 216 countries with confirmed cases of 16,775,633

repository, so it will only be accessible to people who have access to the newspapers from 15th March to 15th July 2020. However, we have attached a text file in txt format containing covd-19 corpus used for this analysis.

**Funding:** The authors received no specific funding for this work.

**Competing interests:** The authors have declared that no competing interests exist.

and 661,244 deaths as of 29[th] July 2020. The pandemic was not only led to a massive death toll but had also put a serious halt on the political, social, and economic infrastructures of the states [3].

Although microbiologists, epidemiologists, and public health officials had attempted to understand the biology, spread, and best response to Covid-19, the general public also sought to understand the unfamiliar as well as the threatening condition created by Covid-19. In his work on how to encourage the public to take action or accept political decisions in unknown environments, Edelman [4] argued that "people who are anxious and confused are eager to be supplied with an organized political order—including simple explanations of the threats they fear—and with reassurance that the threats are being countered" (p. 65). The alarming condition generated by Covid-19 was precisely the situation where common individuals want a simple and familiar explanation of the threat.

Opinion leaders, government, newspaper editors, etc., play a decisive role in shaping and presenting the issues in society [5]. Newspapers are particularly known to lead in the initiation of discourse to shape the opinions of people [6]. With the outburst of the pandemic, the newspapers have been disseminating the ideological construction of equating Covid-19 with the 'War'. According to Van Dijk [7–10] ideology is a form of social knowledge which is shared by a group. In another way, ideology is deeply rooted in the social representations shared by the members of a group [7]. It consists of "principles that represent what is good or bad for the group" [8] thus maintaining the group's norms and values. The interchange of language and ideology has attracted linguists long ago [9, 11, 12] and lead to the development of the interdisciplinary field of Critical Discourse Analysis (CDA). In recent times, CDA combines with Cognitive Linguistics to form the new field of Critical Metaphor Analysis (CMA) to analyze ideologies that are metaphorically implied in language.

As a linguistic device, metaphor is deeply rooted in the language; it is pervasive in our thought and reflected in our daily language. Lakoff and Johnson [13] claim that metaphor is the understanding of one domain in terms of another through which we seek persuasion and 'to define reality [. . .] through a coherent network of entailments that highlight some features of reality and hide others'. Metaphors are defined as interpersonal devices that facilitate the creation of a relationship with the public, being often used with a persuasive function as Charteris-Black [14], Charteris-Black [15] Ferrari [16] Kitis and Milapides [17] mentioned in their works. Metaphors are thus key linguistic devices for constructing social relations and creating, contesting or legitimating specific social, cultural or political and ideological representations of the world [14, 18, 19] and revealing the underlying ideologies [20, 21]. Due its influential role, metaphors has remained a prevailing topic while discussing Covid-19 disease discourse across various languages and cultures as Zibin [22], Younes and Altakhaineh [23], Castro [24], Semino [25] have mentioned in their studies. The current study will investigate the use of metaphor in Pakistani Newspaper discourse and draw cross cultural and linguistic implications while comparing with the aforementioned studies. The method used in this study is the Critical Metaphor Analysis (CMA) by Charteris-Black [14] an approach that is an amalgamation of CDA, CMT, and corpus linguistics. According to him, a critical analysis of metaphors can be employed to uncover the "covert" and "possibly unconscious" intentions of language users. Therefore, this research paper specifically employs CMA [14] to investigate the metaphorical depiction of 'Covid-19 as War' in the discourse of famous Pakistani newspapers. Through this study, we unveil the role of conceptual metaphors in shaping thoughts and actions and more specifically the role of WAR metaphors in Pakistani newspapers. The research imbibes a critical lens to comprehend the language of Pakistani Newspapers as a persuading tool to shape the thoughts and actions of the general public against the Covid-19 disease. Monoconc Corpus Tool has been utilized to analyze the newspaper discourse on Covid-19 as WAR and the

further entailments of this metaphor, precisely the presentation of BATTLE, ENEMY, FORE-FRONT SOLDIERS, COMBATANTS, and VICTORY.

## Rationale for focusing on the war metaphor

The choice to concentrate on the war metaphor in our analysis of metaphorical representations of Covid-19 in corona discourse in Pakistan stems from several compelling reasons that underscore the significance and relevance of this particular metaphor:

1. **Prominence and Prevalence:** The war metaphor has been consistently pervasive in global discussions surrounding the Covid-19 pandemic. It has been utilized by governments, public health agencies, and media outlets around the world, making it a highly prominent and recognizable metaphor. Given its wide-reaching usage, it is essential to investigate how this metaphor influences public perception and communication strategies within the context of Pakistan.

2. **Metaphor Intensity:** The war metaphor is characterized by its inherent intensity and urgency. When applied to the pandemic, it evokes a sense of crisis, collective effort, and the need for immediate action. Examining this metaphor offers valuable insights into how media outlets in Pakistan emphasize the gravity of the situation and the necessity for a unified response.

3. **Conceptual Clustering:** The war metaphor often forms clusters of related sub-metaphors, including "frontline workers," "battles," and "victory." Analyzing this metaphor in detail allows us to explore the interconnected web of metaphorical expressions surrounding the pandemic, shedding light on the multifaceted nature of this linguistic framing.

4. **Public Perception and Policy Impact:** Metaphors play a significant role in shaping public perception and influencing policy decisions. The war metaphor is frequently associated with government responses, policy measures, and public compliance with safety protocols. By examining this central metaphor, we can better understand its impact on public health communication and the formulation of governmental strategies.

While our primary focus is on the war metaphor, we acknowledge the presence of other metaphorical expressions in the Covid-19 discourse in Pakistani newspapers. These metaphors, such as those related to health, natural disasters, or economics, are undoubtedly significant and contribute to the overall framing of the pandemic.

To address this, we will briefly explore and mention other noteworthy metaphors in the paper. This approach allows us to maintain a comprehensive view of the metaphorical landscape, ensuring that our analysis is both in-depth and inclusive. However, our primary objective is to provide a detailed examination of the war metaphor due to its widespread usage and its profound impact on public understanding and policy decisions. By focusing on the war metaphor while also acknowledging the presence of other metaphors, we aim to offer a well-rounded analysis of how Covid-19 was framed in the context of corona discourse in Pakistan and the multifaceted language used to convey its implications.

## Conceptual metaphor theory and medical discourse

Towards the end of the second millennium the world saw the outbreak of new epidemics, as had not been seen since the Spanish flu pandemic. According to Washer [26], a number of factors, including changes in human demographics and behavior, technology and industry, economic development and land use, international travel and trade, microbial adaptation and

change, a breakdown in public health measures, etc., contributed to this renewed susceptibility to infectious diseases.

It is interesting to note that sociologists, linguists, discourse analysts, and communication scholars have been refocusing their scholarly attention on illness communication in parallel with the various outbreaks of epidemic diseases that occurred in the early 2000s, with each case focusing on a different infection, such as Ebola [27, 28], BSE, or mad cow disease [29], foot and mouth disease [30, 31], SARS [32, 33], avian/bird flu [34–36] swine flu [37], MRSA [38], and Zika [39, 40].

Therefore, it should come as no surprise that the Covi-19 epidemic, the most severe and widespread of recent outbreaks, was accompanied by an unprecedented communication effort in the media, in public discourse on health, and in the political sphere. This effort in turn sparked a wave of new studies on coronavirus communication. Special issues of journals have recently been released or are being prepared, such as the Journal of Psycholinguistic Research's special issue on the communication of Covid- 19 with major emphasis on effect of pandemic on cognition and novel language practices [41]. Another journal named as Multilingua edited by Jie Zhang and Jia Li focuses on the stigmatization of certain languages in crisis communication from an essentialist perspective [42]. Importantly, multiple publications on health issues, language changes and communication modifications were produced in the aftermath of Covid-19. The emphasis is on a number of different factors, such as denominations given to the disease and their political ramifications [43], with some research concentrating on particular languages (such as Arabic: [44] [45]). A fascinating collective study, mostly by authors from Malaysia, considers a wide range of potential linguistic approaches that could be used to investigate COVID-19 communication [46] [47]. The systematic use of metaphors, however, has been so pervasive and intense in COVID-19 communication that it has drawn the majority of scholarly attention from linguists, discourse analysts, anthropologists, sociologists, communication specialists, and researchers in cultural studies.

Through comparisons with simpler notions, metaphors play a significant part in helping complicated and abstract topics be better comprehended [48]. The power of metaphor to frame subjects in certain ways, stressing some characteristics while putting others in the background, is related to its capacity for evaluation [13, 49]. This central aspect of metaphor is known as 'Framing' by which words and phrases are utilized to create a particular way of thinking about a topic or a social interaction" [49]. According to Semino et al. [50] there are three different metaphoric approaches that take these framing effects into account: cognitive, discourse-based, and practice-based. Lakoff and Johnson [13], who explored the way individuals employ metaphors to think about various types of experiences, established the cognitive viewpoint. Meanwhile, the discourse-based approach examine forms and functions of metaphor with specific attention towards who uses them, why, in what circumstances, and with what potential implications and repercussions [51]. The practice approach, on the other hand, examines how metaphors might improve or impair communication in specific institutional contexts (such as healthcare) and specifies which metaphors should be employed and which should be avoided [52]. Nonetheless, all the three perspectives on metaphor help in understanding linguistic practices and creating impacts in communication.

Lakoff and Johnson [13] proposed a two-domain model which contended that conceptual metaphors are the surface manifestations of underlying conceptual relationships between two areas. The conceptual metaphors developed through a systematic mapping between two conceptual domains as the mind works by transferring ideas from one conceptual domain (a source domain) to another (a target domain). A conceptual metaphor is therefore nothing but the realization of conceptual structures in a language [53]. In his work, Knowles [54] explained that conceptual metaphors equate two concepts as in ARGUMENT IS WAR. Here, WAR is

the source domain as the concept is drawn from it; while the ARGUMENT is the target domain as the concept of the source domain is applied to it. However, with time the traditional conceptual metaphor theory of Lakoff and Johnson was criticized basing upon many issues like absence of contextual dimension and restricting to two domains by various researchers. In the given backdrop Kövecses, [55–57] developed extended conceptual metaphor theory that differs from the standard view in two ways; namely, in that it is not only a cognitive theory of metaphor but it has a strong contextual component, and that it views each conceptual metaphor as existing not only on a single level (that of domains or frames) but simultaneously existing on four hierarchical levels of schematicity (those of image schemas, domains, frames, and mental spaces). Kövecses' [57] novel approach and advanced model add conceptual knowledge to metaphorical knowledge and add communicative dimensions of understanding, producing and storing metaphorical meaning. This approach will certainly open new avenues to look at old problems.

## WAR metaphor in discourse

WAR, in general terms, is a hostile contention by means of armed forces of opposing parties, nations and states. The concept of WAR has been used time and gain for different types of human disagreement and has been identified as the source domain that is realized in our everyday conversations and situations. Various military words like Battle, Attack, Combatant, Win, Lose, Victory, etc. are used to define non-military situations as well. WAR metaphors have been employed in various discourses. Lakoff and Johnson [13] introduced the concept of ARGUMENT IS WAR in their research study. Here, WAR is the military source domain as the concept is drawn from it, while the ARGUMENT is referred to as the target domain. Koller [23] examined that WAR as a source domain is pervasively employed in the business and financial discourse. In the same manner, Chung [58] explored the conceptual metaphor of ECONOMY IS WAR in English and Chinese Newspapers. Charteris-Black [14] also explored the conceptual metaphors of TERRORISM IS WAR in the aftermath speech of 9/11 by Bush and quoted him as saying, "we stand together to win the war against terrorism". Moreover, while exploring the disease SARS, Chiang and Duann [59] found the conceptual metaphor and naming strategies for SARS in three major newspapers named as The United Daily News in Taiwan, The Liberty Times, and The People's Daily in China. The researcher focused on the DISEASE IS WAR metaphor as constructing the concepts of Self and Others in the audience.

While the majority of authors oppose the use of WAR (or "military" or "martial") metaphors in patient care, Hauser and Schwarz [60] in particular argued that the use of "bellicose" metaphors for cancer patients does not have any positive effect because they increase fatalistic beliefs about cancer prevention, failing to motivate people to immediately see their doctor when they have any signs of cancer. As a result, it is recommended that when speaking with cancer patients, WAR metaphors should be avoided and positive metaphors like JOURNEY should be followed [52]. Later on, Chircop, & Scerri (2018) [61] argued that usage of metaphors for discussing cancer should not be discouraged as metaphors could create sense of empowerment and disempowerment depending upon individual's perspective.

In a corpus-based study looking at online forum posts by 56 different contributors to a publicly available UK-based website for people with cancer Semino, Demjén and Demmen [62] confirmed that recourse to conceptual metaphors involving WAR as a source domain is most frequent, followed by JOURNEY metaphors, which however are considered less problematic by experts. The results of their research showed that this use of WAR metaphors is not always negative, but can have contrasting effects depending on the context and the way they are used. In particular, recourse to them can be useful or detrimental as a function of patients' degree of empowerment, i.e. their degree of agency and control of events, or disempowerment,

especially in cases of failure to recover where violence metaphors end up inducing guilt and frustration for something for which patients are certainly not responsible [62].

Communication about COVID-19 has been marked by a preference for WAR metaphors since the disease first appeared on the global stage, when little was known about its characteristics and impacts. The same critiques that had previously been leveled at its usage for cancer eventually followed this preference, with interventions from medical professionals who criticized its disadvantages and potential negative effects [63, 64]. Given that battles are fought but can also be lost, these criticisms claim that while WAR metaphors may be useful to communicate certain aspects because they may resonate with the general public and promote a "all in this together" mentality, they can also have seriously detrimental effects and often lead to feelings of disempowerment, blame, and despair. WAR metaphors are useful to communicate the severity of health crisis among public but may generate panic and fear and sense of defeat among citizens [65] [66]. The assumption that we are battling an enemy virus invasion also encourages the medicalization of prejudice, which stigmatizes minorities [67]. Furthermore, by portraying the pandemic as an unexpected emergency, governments who have neglected to implement effective prevention and response measures are no longer held accountable [65] Rohela et al. [68] contended that our pandemic response has become naive by referring to it as a "war." It has further harmed social cohesion in general society, stigmatizing people, causing a rift between various healthcare system participants and sectors, and raising the possibility of a further decline in the doctor-patient relationship. The metaphorical depiction of the COVID-19 situation as a war may be harmful because it may change how people perceive and respond to the epidemic, encouraging support for authoritarianism and restrictions on civil freedoms among the populace [69]. Another disadvantage is that political leaders may use the notion of a nation at war as justification for excessively authoritarian policies and the appropriation of vast and extraordinary powers [65]. Moreover, Musu [70] and Schwobel-Patel [71] contended that such language practices promotes nationalism instead of internationalism and global cohesion towards pandemic. Likewise, Garzone, [72] contended that high frequency of war-related metaphorical expressions suggests that they have now become conventional and lost their resonance, thus reducing their potential impact. However, the findings of Benzi & Novarese [73] were in contrast with the earlier studies as they address the social consequences of utilizing WAR metaphor, and argued that such metaphors do not restrict citizens to accept limited civil liberties and authoritarian policies. Chapman and Miller [74] also discuss the positive implications of using war-like language for Covid-19 pandemic and concluded that such vocabulary could be used to refocus public attention on external threats while seeking to advance a policy or political agenda.

Against this background, this study looks at the use of metaphors in discourses on Covid-19 in Pakistani Newspapers in order to answer the following research questions:

1. How the language of Pakistani Newspapers has inculcated the conceptual metaphor of Covid-19 as WAR?

2. How do the Pakistani Newspapers use the military concept of WAR to create SELF Vs OTHER distinctions between the Pakistani people and the medical illness of Covid-19?

3. What are the ways through which the conceptual metaphor creates urgency and mobilization among the public regarding Covid-19?

## Method and material

The following section introduces the theoretical approach of Charteris-Black's [14] Critical Metaphor Analysis (CMA) as well as the methodological framework adopted for the analysis

of data to achieve the research objectives of the study. This approach is of immense importance as the theoretical concepts provide a guideline for the study of underlying philosophical assumptions and research strategy underpinning the selected methodology and data collection techniques.

Charteris-Black's [14] Critical Metaphor Analysis (CMA) is an original, multi-layered model which integrates Conceptual Metaphor Theory, Critical Discourse Analysis, and Corpus methodology to study the conceptual metaphors inculcated in a text. This model aims to uncover the hidden (and possibly unconscious) intentions of language users behind the usage of metaphorical expressions [14]. He expresses his views that metaphors are present at the thought level whereas express at the language level. Therefore, CMA could be best utilized to detect the presence of conceptual metaphors in the language used in the society. According to Charteris-Black [14], the replacement of literal expressions with the metaphorical one is not only done to achieve composition or style but also to add persuasion to the language. Conceptual metaphors are largely inculcated in a context to influence the opinion as well as the perception of the general public regarding any social or political issue. In his work, for the comprehensive analysis of metaphors, Charteris-Black [14] included all three components of linguistic, cognitive, and pragmatic criteria, since any one of these components cannot alone provide an inclusive explanation of metaphorical expression. Thus, Charteris-Black [14] gives an analytical framework to study the metaphors, their usage, and the likely effects produced by them in a text.

In his model, Charteris-Black [14] explains three stages of metaphor analysis: identification, interpretation, and explanation.

## Metaphor identification

The first stage of metaphor identification by Charteris-Black [14] includes two steps: (1) A close and cautious reading of the collected data for identification of candidate conceptual metaphors. After this identification, metaphor keywords are categorized and selected by the researcher for further analysis in the corpus. (2) After the first step, a qualitative analysis is carried out to examine the context and to decide whether each usage of the keyword is metaphorical or literal in nature.

## Metaphor interpretation

This stage involves 'establishing a relationship between metaphors and the cognitive and pragmatic factors that determine them' [14]. The types of social relations constructed through the metaphors are interpreted in this phase.

## Metaphor explanation

According to Charteris-Black [14], metaphor explanation involves the way metaphors are interrelated to the context in which they exist. In this phase, a researcher can uncover ideological and rhetorical motivations through the identification of the discursive function of the metaphor. Careful analysis of the corpus, in which metaphors occur, makes it possible to find evidence of the motivation of the text producer. In the explanation part, the construction of Self and Other is thoroughly studied for explaining the motivations behind the ideological use of metaphors.

The data collected for this research paper included the editorials of three renowned Pakistani English Newspapers i.e. Dawn, The Express Tribune, and The News (List of Top Pakistani newspapers, 2020). These three newspapers were chosen due to the fact of their leading role, widespread audience, and mass circulation throughout the country. The editorials

selected for the study ranged from the period of 15th March to 15th July 2020 as it was the peak time of the Covid-19 outbreak in Pakistan. Furthermore, fifty (50) editorials on the topic of Covid-19 were selected for the study from the newspapers.

Following the mixed-method approach, the data is analyzed using both the methods of quantitative and qualitative research analysis. For the quantitative analysis, a specialized corpus containing the selected Pakistani Newspaper editorials is constructed and named as PakN-Covid-19. The size of the corpus is 17621 words. Moreover, Monoconc Corpus Tool is utilized to analyze the metaphorical depiction of Covid-19 as a WAR in Pakistani Newspaper discourse. The details of corpus are mentioned below in Table 1:

In addition to it, the study inculcates the careful reading of collected data to identify 'candidate conceptual metaphors' that categorize commonly used words with metaphorical meaning as keywords for metaphors. Following Charteris-Black [14], who selected keywords from the source domain, we also selected the keywords from the source domain of WAR. A manual analysis of the data set was conducted to identify metaphorical expressions describing the illness experience of the participants. Metaphorical expressions were identified using the well-established Pragglejaz Method for Finding Metaphorically Used Words [75]. This method is described in the following Table 2.

Once the metaphorical expressions had been identified in the data set, each metaphorical expression was allocated to a semantic field corresponding to its literal meanings. We selected the following metaphorical entailments of WAR as keywords to search in PakNCovid-19: 1) BATTLE, 2) ENEMY, 3) FRONTLINE SOLDIERS, 4) FIGHT, 5) VICTORY. After the location of the keyword, we thoroughly examined the context (i.e. the preceding and following sentences) for the sole purpose of deciding the metaphorical depiction of the keyword in that instance. In the qualitative analysis part, the concordances of keywords were interpreted and explained with a critical voice inherited by CMA.

This research is a corpus-driven study as inculcating corpus methodology to examine conceptual metaphors is a novel and significant approach that aids in reducing and limiting

**Table 1. Size of PakNCovid-19 corpus.**

| Sr. No | Name of Newspaper | Month | No.of Editorials | Word Count |
|---|---|---|---|---|
| 1. | Dawn | March | 2 | 881 |
| | | April | 4 | 1281 |
| | | May | 7 | 2376 |
| | | June | 3 | 1108 |
| | | July | 5 | 1809 |
| | | **Size of Corpus** | **21** | **7,455** |
| 2. | The Express Tribune | March | 1 | 368 |
| | | April | 3 | 921 |
| | | May | 4 | 1288 |
| | | June | 2 | 741 |
| | | July | 3 | 1241 |
| | | **Size of Corpus** | **13** | **3,959** |
| 3. | The News | March | 1 | 421 |
| | | April | 4 | 1589 |
| | | May | 5 | 1934 |
| | | June | 4 | 1367 |
| | | July | 2 | 896 |
| | | **Size of Corpus** | **16** | **6,207** |
| | **Total** | **Size of Corpus** | **50** | **17,621** |

**Table 2. The steps involved in the Pragglejaz Method for Finding Metaphorically Used Words.**

| Step 1 –Decide about the boundaries of words | The excerpt 'doctors are frontline soldiers' does not literally mean that they are military persons |
|---|---|
| Step 2 –Establish the contextual meaning of the word being examined | The contextual meaning of 'frontline soldiers' is to make doctors feel about their crucial duty to save others and be combatants |
| Step 3 –Determine the basic meaning of the word | The front line soldiers are the soldiers of opposing armies who are facing each other and where fighting is going on. [. . .] (Collins, 2023) |
| Step 4 –Decide whether the basic meaning of the word is distinct from its contextual meaning | The basic and contextual meanings of 'frontline soldiers' are distinct |
| Step 5 –Decide whether there is some form of similarity between contextual meaning of the word that can be related to its basic meaning. | During Covid-19, the doctors have the major responsibility of saving lives, treating infected people, and disseminating precautionary measures, and are known to be saviors of country. Similarly, soldiers save their country in a war |

researchers' prejudices. Instead of restricting to one article, our research entails the study of a large number of articles which in turn reduces the analysts' cognitive biases and provide an objective and less selective data. A large data set has an increased capacity of showing "general patterns and trends" [77] and "large and systematic" evidence of language use. The use of corpus data rather enables the recognition of more frequent speech patterns [24, 76, 77]. Moreover, the corpus examination of language usage can determine how typical the discourse may be. The frequency of occurrence, collocations, and cluster formation in the data reveals certain repeated patterns in the corpus, hidden ideologies, and power differences in the language. In examining conceptual metaphors, the employment of corpus methodology is effective as it includes the perspective of critical discourse analysis [23, 24, 78]. Empirical evidence for socio-cognitive approaches to metaphor could be supplemented through corpus data. In this study, a corpus-driven metaphor approach is followed as it allows the triangulation of data [76] and increases the systematicity [78] and productivity [14] of research. The qualitative analysis of corpus data allowed us to have a closer and more insightful analysis of Newspaper language. Moreover, having electronically stored natural data and computer tools to analyze that data provide accuracy and authenticity to research.

## Researchers and collaborative analysis

The analysis of the corpus in this study involved a team of four researchers. This collaborative approach was chosen to ensure a comprehensive examination of the data and to enhance the reliability and validity of the findings. The collective expertise of the research team allowed for a multifaceted exploration of the metaphors present in the text.

Following the identification, interpretation, and explanation stages of Charteris-Black's Critical Metaphor Analysis (CMA), the research team recognized the importance of reaching an agreement to measure the consistency in identifying and interpreting metaphors within the corpus. To address this concern, a consensus was established after individual analyses were performed. Each researcher independently conducted the initial analysis to identify and interpret metaphors in the corpus. Subsequently, the researchers convened to compare and discuss their findings.

## Collective examination and consensus

During these group discussions, any disparities in metaphor identification and interpretation were thoroughly examined and discussed. This collaborative process allowed the research

team to address doubtful cases, resolve differences, refine the analysis criteria, and reach a consensus on the presence and meaning of metaphors within the corpus. The agreement among the researchers was established through a careful consideration of individual analyses and collective decision-making, ensuring a robust and reliable analysis.

## Data analysis

This section presents the data analysis of WAR metaphors in three renowned English Newspapers of Pakistan: Dawn, The News, and The Express Tribune. Fifty (50) editorials on the subject of Covid-19 were specifically chosen and their language was meticulously observed by making a specialized Corpus PakNCovid-19. With the help of the Monoconc Corpus Tool, the source domain of WAR and its entailments were searched. The Table 3 shows source domain and target domain of War metaphors of Covid-19 discourse in Pakistan. It also includes metaphor formulas for the source domains and explains conceptual metaphors related to Covid-19 in the corpus:

The Table 3 provides a focused analysis of key metaphors related to the battle against Covid-19, emphasizing control, confrontation, dedication, opposition, and the quest for victory. It provides a more detailed and structured analysis of metaphorical representations of Covid-19 in Pakistani newspapers. It delves into the intricacies of how Covid-19 was metaphorically represented. By providing metaphor formulas, it explicitly states the relationship between the source domain (e.g., war, health) and the target domain (e.g., control, pandemic). The metaphor formulas in the Table 3 serve as concise expressions that capture the essence of each metaphor. For instance, "Covid-19 IS A WAR" encapsulates the idea that the pandemic was metaphorically seen as a battle or war against an adversary. These formulas provide a clear, at-a-glance understanding of each metaphor used in the discourse. It includes a column that provides explanations for the conceptual metaphors. These explanations offer a deeper understanding of why a specific metaphor was used, what it signified, and how it influenced the discourse related to Covid-19. For instance, the "COVID-19 IS A WAR" metaphor implies that the response to the pandemic involved strategies to control, achieve victory, and combat the virus as if it was an enemy. In short, the Table 3 enhances the analysis of metaphorical representations by providing a systematic and detailed breakdown of the metaphors found in Pakistani newspapers' coverage of Covid-19. It accomplishes this by offering metaphor formulas and explanations, making it an invaluable resource for understanding the language and rhetoric used in the media's portrayal of the pandemic.

Furthermore, each WAR entailment is elaborately discussed as follows:

**Table 3. War metaphors of Covid-19 discourse in Pakistan.**

| Source Domain | Target Domain | Metaphor Formula | Explanation of Conceptual Metaphors |
|---|---|---|---|
| **War against Covid-19** | Control, Victory | WAR AGAINST COVID-19 | The pandemic is framed as a war, emphasizing the need for control and achieving victory. |
| **Covid-19 as an enemy** | Confrontation | COVID-19 AS AN ENEMY | Covid-19 is metaphorically portrayed as an adversary that needs to be confronted and overcome. |
| **Battle against Covid-19** | Challenge, Victory | BATTLE AGAINST COVID-19 | The response to Covid-19 is likened to a battle involving challenges and the pursuit of victory. |
| **Doctors as soldiers** | Dedication, Defense | DOCTORS AS SOLDIERS | Doctors are depicted as dedicated soldiers on the frontlines, defending against the pandemic. |
| **Fight against Covid-19** | Opposition, Triumph | FIGHT AGAINST COVID-19 | The response to the pandemic is framed as a fight against opposition, with the goal of achieving triumph. |
| **Victory over Covid-19** | Achievement, Success | VICTORY OVER COVID-19 | The objective is to achieve victory over the pandemic, signifying success and overcoming challenges. |

## War against Covid-19

War is a military confrontation between two or more geopolitical areas or organizations whose members are geopolitical areas' (SUMO: 'War'), superordinated by Violent contest', 'contest', 'social interaction', Intentional process', 'process', 'physical' and ultimately 'entity' (Fig 1). Because a violent contest necessarily involves two or more parties, it may be inferred that conflict/ opposition exists between them. As each party needs to evoke solidarity among its members to fight and win against an opponent, we propose that the construction of Self and Other serves as an effective strategy for accomplishing this purpose.

According to Lakoff and Johnson [13] conceptual metaphors are utilized to make a highly complicated and unfamiliar concept more familiar to the audience. In the same manner, Pakistani Newspapers had employed the familiar concept of WAR to explain the unacquainted Covid-19 to their readers. The language of Covid-19 articles showed the explicit deployment of military concepts like BATTLE, ENEMY, WAR, SOLDIERS, FIGHT, and VICTORY in stating the situation of the virus in the country. Combining all these metaphors, the Pakistani Newspapers created a SELF Vs OTHER distinction between the audience and the medical illness of Covid-19. The conscious engagement of military concepts regards Covid-19 as an 'alien', 'outsider', and 'enemy', which as result, creates a sense of urgency among the Pakistani

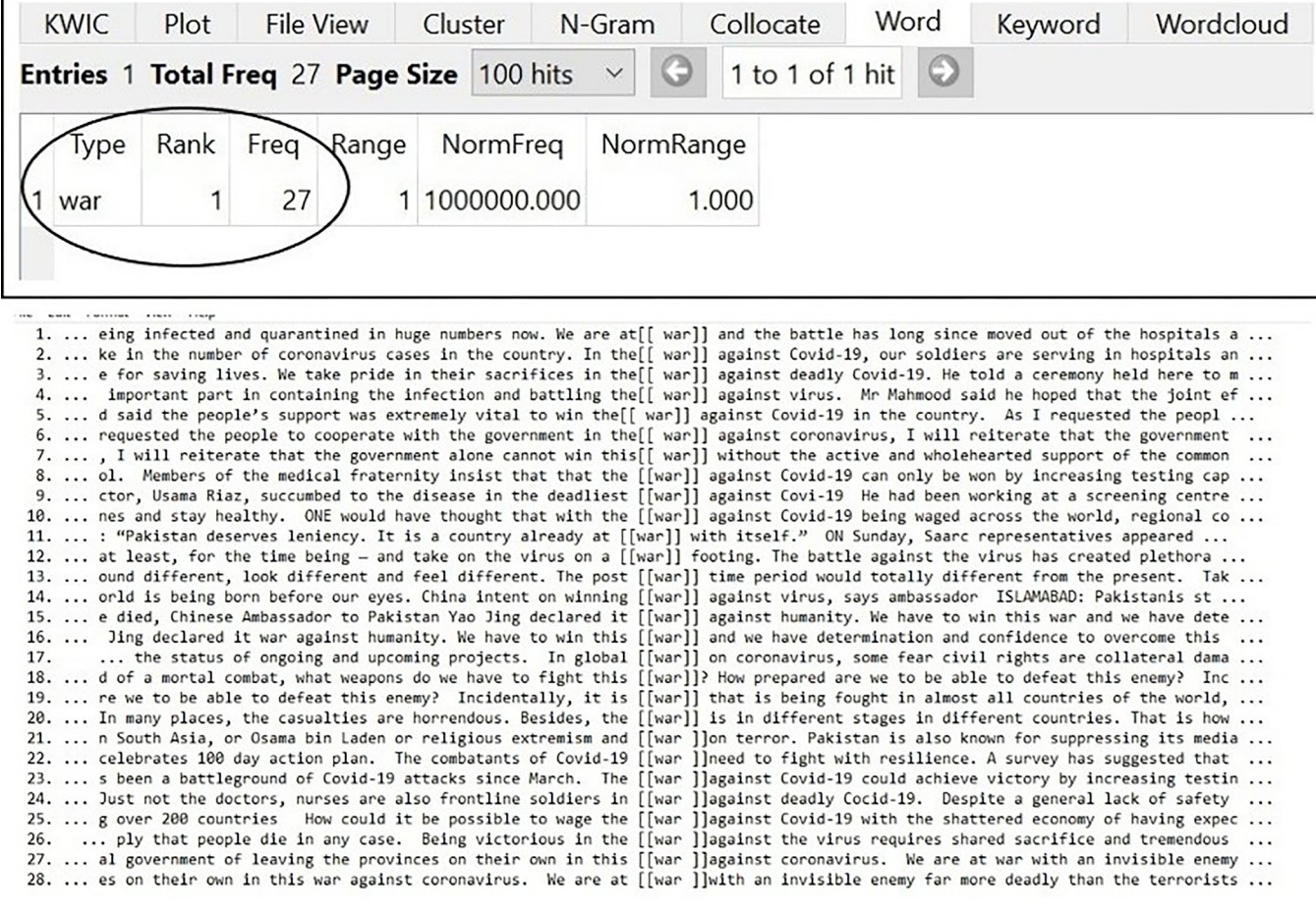

**Fig 1. WAR against Covid-19.**

public and makes them antagonistic to the virus. Following are the hits received for the keyword 'WAR' in the corpus PakNCovid-19: The frequency of the word 'war' has observed in Fig 1 as:

According to Charteris-Black [14] the replacement of literal expressions with the metaphorical one is not only done to achieve composition or style but also to add persuasion to the language. Instead of being called as a medical emergency, the conceptual metaphor of WAR (lines 1–28) has been deliberately utilized by opinion makers to influence the public and to inflict a sense of urgency on them. The Newspapers reiterated the conception that we (the country) were at war against Covid-19, and to win this war we need determination, resilience, and shared sacrifice (lines 16, 22, 26). The metaphorical representation of language creates a distinction between Self (country) and Other (Covid-19) and generates the 'sharply opposed, polarized, binary extremes—good/bad, civilized/primitive, ugly/excessively attractive, [. . .]' variations as suggested by Hall [78]. To inflict negative emotions among the public, the war is called as the deadliest war against humanity which was being fought in almost all counties around the globe (lines 9, 19). Therefore, expressions like these create a generalized hatred against Covid-19 and construct a boundary between the populace and the pandemic.

## Covid-19 as an enemy

In Pakistani Newspapers, Covid-19 had been consciously represented as an 'ENEMY'. As a military concept, an enemy is likely to be met with hatred, vehemence, battle, and war. Therefore, the conceptual metaphor of ENEMY serves as a powerful rhetorical device in shaping the antagonistic thoughts and actions of the public regarding Covid-19. The language in Newspapers constructed the Pakistani government, the populace, the media, and the medical personnel as an in-group while Covid-19 alone was an out-group enemy. This grouping of Covid-19 as an outsider force perpetrated the general public to encounter it with an opposing behavior. When analyzed, the following hits were obtained from the data of Newspaper editorials entitled Covid-19 as an ENEMY. The word "enemy" has the frequency of 20 as shown in Fig 2:

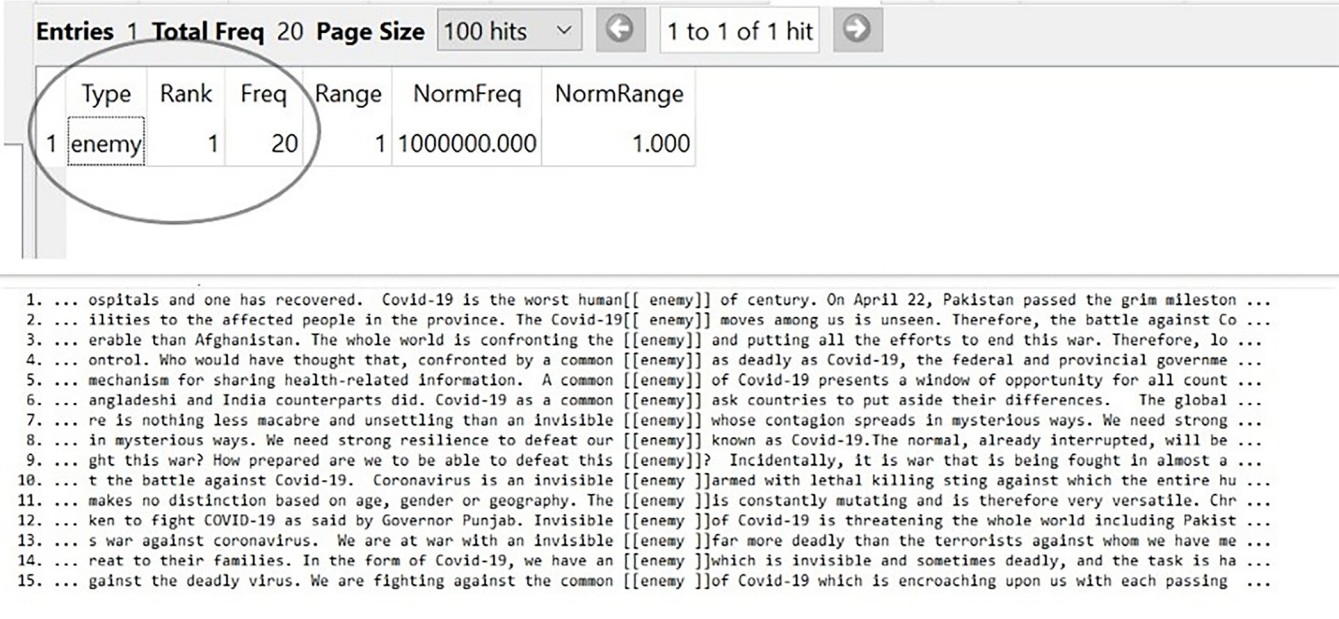

**Fig 2. Covid-19 as an ENEMY.**

The above-given hits, obtained for the conceptual metaphor of Covid-19 is ENEMY, show a trend of depicting Covid-19 through military vocabulary. In past, with the rapid spread of pandemic, the newspapers disseminated the ideology of seeing the virus as an enemy force among the general audience. According to Van Dijk [10] ideology is a form of social knowledge which is shared by a group and conceptual metaphors are derived from such ideological and rhetorical motivations. News media had deliberately used the expression of 'common' with the metaphor ENEMY (as in lines 4, 5, 6, and 15) to reinforce the idea that the whole world is one unit that is encountered with the 'common' threat. In order to maximize the danger posed by the ENEMY, Covid-19 has also been labeled as an 'invisible' ENEMY (as in lines 7, 10, 12, and 13). Despite knowing that the virus was a microscopic bundle of inanimate genetic material, it had been depicted as an ENEMY which was covert, suspicious, and can deceitfully intrude among human beings. Hence, through the application of conceptual metaphors, not only the alarming condition was initiated but also an impulse for defeating the Covid-19 ENEMY was followed (as in lines 8 and 9).

### Battle against covid-19

Due to the outbreak of Covid-19, states, economies, and individuals had faced multifarious issues. The pandemic had wreaked havoc on the political, social, and economic infrastructures of the countries and the public [3]. Amid the outbreak of the virus, people had been remained quarantined; shops and markets had been closed; educational and other institutions had been barred, and flights had been blocked. In that deteriorating condition, people were conceived to be entering into a battle against Covid-19. The conceptual metaphor of BATTLE was consciously employed for persuading the audience to take precautionary measures against the Covid-19. The frequency of the word "battle" is present in Fig 3 as:

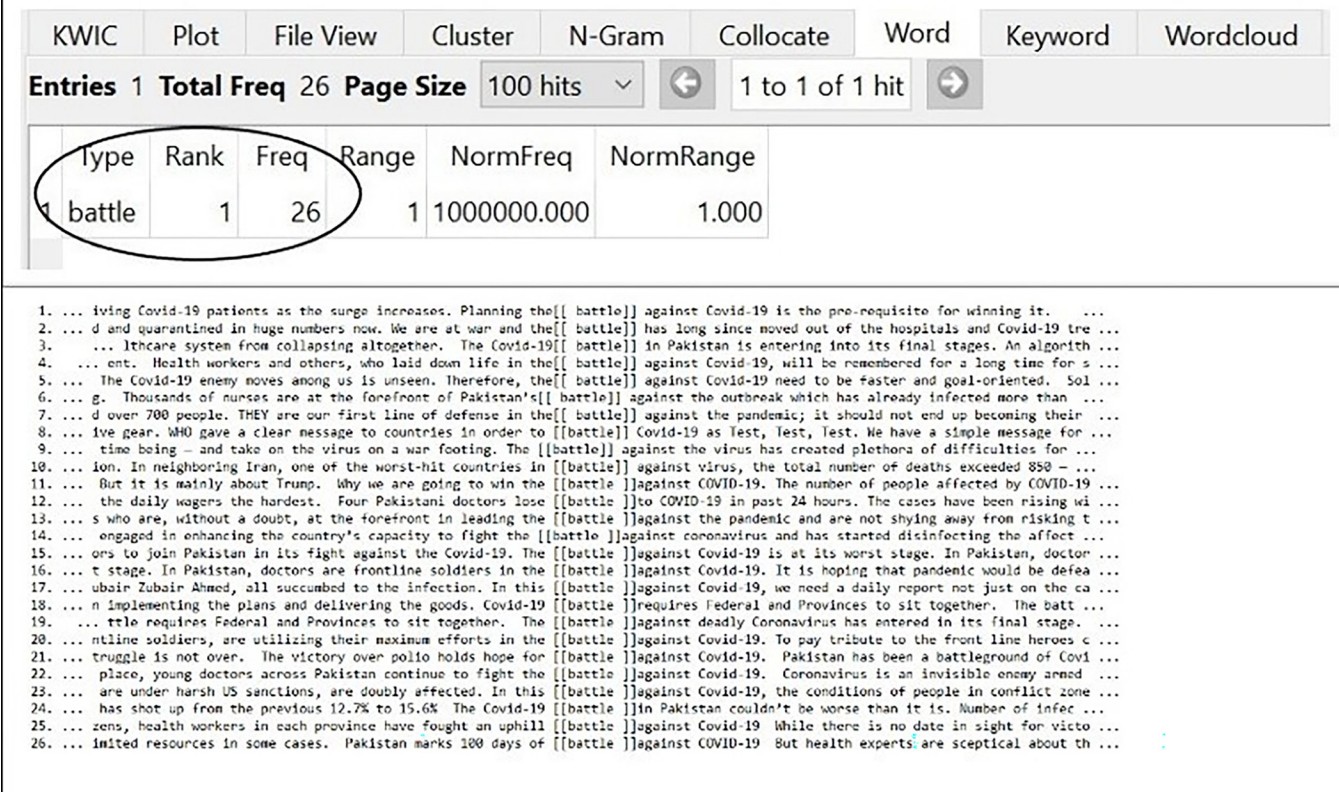

**Fig 3. Battle against Covid-19.**

A battle is generally defined as a sustained fight between large organized armed forces. However, the Pakistani Newspaper discourse showed the cognition of the new concept of Covid-19 through the use of the old idea of battle among the opposing forces. As in cognitive linguists, metaphors are not viewed as decorative or marginal devices, but as a tool for understanding a phenomenon and an embedded intention. In the same manner, the conceptual metaphor of BATTLE shows that Pakistan was combatting and struggling hard against the deadly virus (as in lines 1–26). In this bleak time of the pandemic, the Newspapers seem to be creating anxiety and fear among the audience. The fretful metaphorical representation dispersed negative thoughts and emotions against the opposite party (Covid-19); which in turn, aided in mobilizing the public and channelizing available resources.

## Doctors as soldiers

As the world is complex and unpredictable which offers unexpected calamities and challenges like plagues, floods, and earthquakes to human beings. In order to understand the novel experiences, people resort to familiar explanations and conceptions; thus, forming conceptual metaphors. Charteris-Black [14] expresses his views that metaphors are present at the thought level whereas express at the language level. In the WAR against Covid-19, Newspapers had presented doctors, nurses, and medical staff as soldiers who sacrificed their lives for saving the world from pandemic. The metaphorical depiction of doctors as SOLDIERS in Pakistani Newspapers is obtained through the following hits given in Fig 4:

Soldiers are the heroic figures for a nation. They are the ones who fight as part of an army and serve the national interest of a country. They selflessly take part in a war and do fighting in planes, on the grounds, or from boats. As ferocious Covid-19 had enveloped Pakistan, with hospitals brimming, doctors dying and infections escalating at an insurmountable rate, the Pakistani Newspapers had portrayed doctors as the SOLDIERS of the country who have been gallantly sacrificing their lives for saving others. The conceptual metaphor of SOLDIERS exhibits that Doctors were playing an indispensable role in fighting against Covid-19. They had been called as 'FRONTLINE SOLDIERS' (as in lines 2, 4, 5, 6, 7) which augments their already substantial role in the pandemic. Their 'leading role', 'indefatigable efforts', and 'defense' (as in lines 2, 4, and 7) against the virus depicted them as patriotic, responsible, and

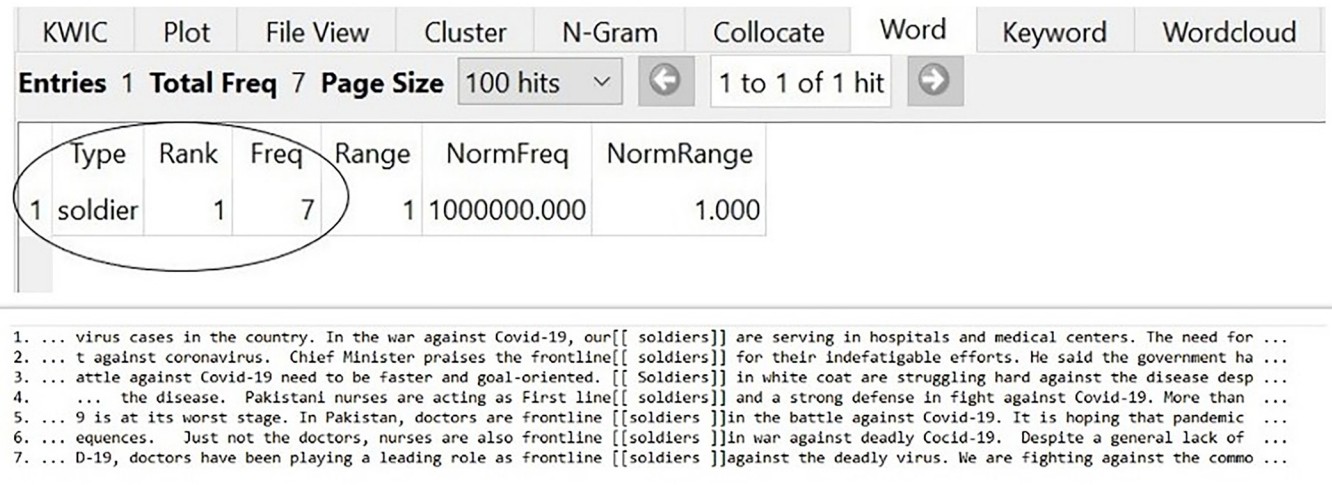

**Fig 4. Doctors as SOLDIERS.**

united members of society. Collectively, these linguistic terms transfigure medical profession-als from citizens into SOLDIERS. The metaphorical representation of the Doctors' role shows their inclusiveness in Self and derivation from Others (Covid-19), which is one of the focal purposes of metaphors as described by Charteris-Black [14].

## Fight against Covid-19

Another important conceptual metaphor used in Pakistani Newspaper, belonging to military source, was the FIGHT against Covid-19. The collected data showed 25 metaphorical hits of the keyword 'FIGHT" when searched through the corpus tool as in Fig 5. The excessive use of this particular metaphor elucidates that the country was in a WAR condition where we (the country) had to FIGHT against the other (Covid-19).

Fight is a military term, particular use when someone has to contend in battle or to combat physically. Despite this, the struggle against Covid-19 was not a physical combatting with weapons, but Newspapers had been seen as constructing the reality that country was at WAR against the disease. This conceptual metaphor reflects deeply held patterns of opinion makers about the crisis. By reaching the cognition of the public, the Newspapers had been seen to influence their thoughts and actions regarding the maintenance of social distancing and the adoption of preventive measures against the disease. The Newspapers dispersed the ideology that the ammunition used in this FIGHT were the masks, gloves, and sanitizers, and people should revert to them. Another important purpose of depicting the FIGHT against the virus is to enhance national unity and to build a national front (as in lines 1, 2, 9, 15, and 18). To create

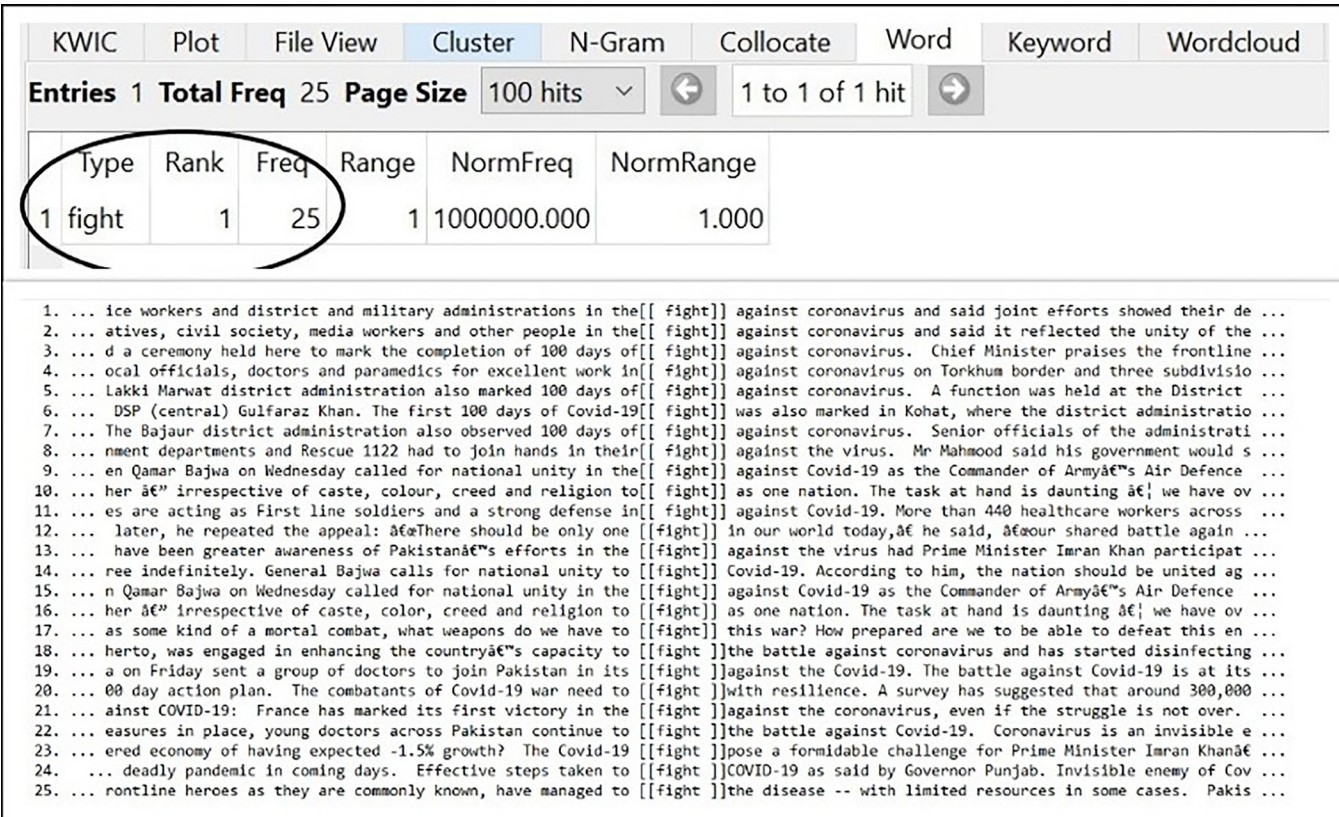

**Fig 5. FIGHT against Covid-19.**

a strong fence against the deadly virus, the law enforcement agencies, army, doctors, social activists, media personnel, and the civil society were called as COMBATANTS (line 20) of this FIGHT. Therefore, it is evident that the conceptual metaphor of FIGHT had widened the gap between the Pakistani people and Covid-19 (an alien agent).

## Victory over Covid-19

The BATTLE against the ENEMY, the actions of SOLDIERS on the FRONTLINE, and the general public as the COMBATANTS required one final step to complete the WAR metaphor and that is VICTORY against Covid-19. It might seem obvious that to eliminate Covid-19 from Pakistan, VICTORY was the desired dream as put forward by Pakistani Newspapers. The collected data showed the following 11 hits of VICTORY in the selected Newspapers' language as in Fig 6:

In the given data, by observing the prospect that the country would FIGHT until it WINS with VICTORY, the Pakistani Newspapers fulfilled the requirement that the WAR against Covid-19 will end in VICTORY (as shown in lines 1–11). The HOPE of DEFEATING the pandemic had been raised by the opinion makers (lines 2, 5) which has the capacity to entail the expectations of the public. The metaphor of VICTORY has seen to be influencing people in foreseeing the normalcy and returning to pre-war conditions; thus, achieving the basic purpose of conceptual metaphors in language. Conceptual matters, as a matter of fact, express speakers' intentions and conduct ideological influence on the audience [17].

## Discussion

The outbreak of Covid-19 had been one of the most devastating events in recent memory. It all started with a mysterious pneumonia cluster in China in late 2019, which was reported to the World Health Organization (WHO). In a matter of weeks, the virus had spread to Iran, Italy, and South Korea, prompting the WHO to declare a global pandemic on March 11th, 2020. The USA soon followed with lockdown and travel restrictions to New York [47]. It would not be wrong to say that the virus had drastically changed the way people live, work, and interact with one another, and had caused immense hardship and suffering around the world.

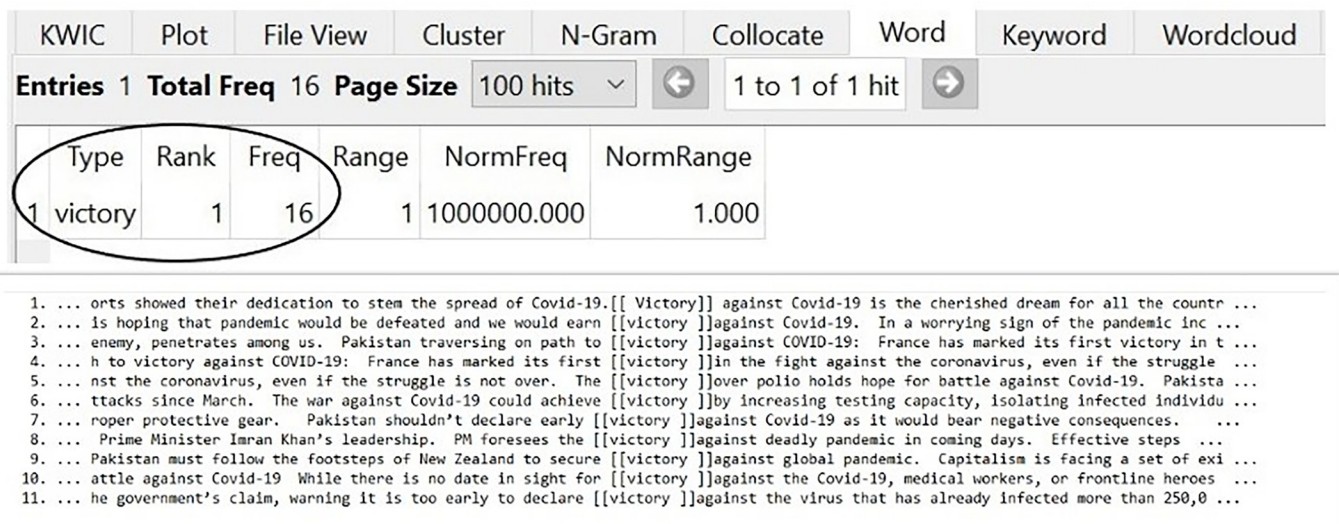

**Fig 6. VICTORY over Covid-19.**

The use of the war metaphor to describe the Covid-19 pandemic has been pervasive. From President Trump's invocation of the Defense Production Act to the deployment of United States military resources, the fight against Covid-19 has been framed as a battle of good versus evil. Although this terminology is often used for dramatic effect, it can have serious implications. This kind of language can give the impression that the virus was an enemy that needs to be vanquished, rather than a crisis that must be managed. It can also lead to a false sense of security, as if the conflict can be won with military might or a single decisive action. In reality, the fight against Covid-19 required a multi-pronged effort and a sustained effort over time. Ultimately, it is important to be mindful of the language we use when discussing the virus. The war metaphor can be useful in some cases, but it is important to remember that the virus is a serious public health crisis that requires a coordinated response from governments, healthcare providers, and individuals Many studies have reported the use of military forces to combat Covid-19 [79–83].

The War on Covid-19 had taken a significant toll on the nation, both economically and socially. The financial burden had been substantial, while the psychological toll had been even greater. The social and political unrest that had been stirred up by the pandemic had left many feeling uncertain and scared. Furthermore, the century's deadliest pandemic had brought to light the immense racial disparities that exist in social hierarchy. This is a major cause of concern as it reflects the unequal access to healthcare and resources for certain communities. The racial divide in Covid-19 cases, which was disproportionately impacting communities of color, was a major indicator of the deep-seated inequality that exists in our society. This has led to a heightened awareness of the need to address these inequalities and provide equal access to healthcare and other resources. The government, private organizations, and citizens must all take action to ensure that these disparities do not persist. Only then can we ensure that all citizens have equal access to justice and healthcare [19]. Likewise, in Pakistan, Rana, Ayoub and Ghilzai [20] noticed that individuals with fewer financial means were subject to social injustice and had difficulties accessing the Covid-19 vaccine due to health inequity.

Previous studies have reported inequalities in health care services in their respective countries as Zibin [22], Younes and Altakhaineh [23] reported the adverse conditions in Jordanian context. Similarly, South Asian countries like Pakistan also witnessed the same effects where the wealthier population had access to better health care services than the poor [84–87]. In order to address the inequities in health care, it is essential to remove the barriers that prevent people from having equal opportunities to health care resources. This can be done by providing adequate health care services to all citizens regardless of their financial status. Additionally, the government should invest in health care infrastructure, including building hospitals and clinics, and providing health care subsidies to low-income households. Furthermore, raising awareness of the importance of preventive health care, such as regular checkups and vaccinations, can also help to reduce health disparities.

Metaphor is an effective tool for communication and thinking, as it can be used to bring clarity and understanding to complex topics. However, there is a debate around the world, about which metaphors are suitable and effective and the alternative metaphors that can be used. Semino [50] suggests that Fire metaphors have been considered very unique and appropriate as compared to War metaphors. However, Fire metaphors may not be the best option to depict virus, and may not affect people who are not afraid of fire [60]. Fire metaphors may be inappropriate. Consequently, it is critical to have a well-educated and context-sensitive approach when selecting metaphors for public health communications. This will help to present an appropriate and effective message to different audiences. While metaphor can be a useful tool for communication, it is important to consider the context and select the most appropriate metaphor, as it can help to effectively convey public health messaging.

Instead of other metaphors, the metaphor of war is being employed frequently to help create a sense of solidarity in the fight against adversaries as mentioned earlier by Semino [25]. In the light of the above discussion the present study suggests that this metaphor has become a powerful tool for creating a sense of urgency and unity amongst those who are fighting for a cause. It has been used to bring stability and calm to situations of social upheaval, providing a sense of purpose in the midst of chaos. However, the analysis of this metaphor has uncovered a more complex picture—one that oversimplifies the risks of personal sacrifice as a shared sacrifice [65–67]. The metaphor of war is often used to create a sense of solidarity and unity amongst those who are fighting for a cause. The "war" is seen as a battle against a common enemy, which allows people to focus their energies on one unified goal. This metaphor is particularly powerful in times of social upheaval, providing a focus for people who are facing uncertain times. By framing the situation as a battle against an enemy, it helps to bring a sense of purpose to a chaotic situation. However, there are some drawbacks to the use of the metaphor of war. By oversimplifying complex social issues, it can reduce them to a simple dichotomous conflict. This oversimplification can lead to a false sense of security, as it glosses over the complexities of the situation as indicated by Musu [70] and Garzone [72]. It also ignores the personal risks associated with a conflict, presenting them as a shared sacrifice. The recurring use of war metaphors loss its originality and familiarity with disease as Philip [88] points out, "Repetition and reuse of a metaphor lessen its impact".

Overall, this paper has explored the implications of the metaphor of war and its oversimplification of complex issues. The use of the metaphor of war is a powerful tool for creating a sense of urgency and unity amongst those who are fighting for a cause. It can bring a sense of purpose to chaotic situations, as well as providing a focus for those who are facing uncertain times. However, the analysis of this metaphor has revealed a more complex picture—one that oversimplifies the risks of personal sacrifice as a shared sacrifice. The use of war metaphors can also be seen as a form of propaganda, as it can be used to manipulate public opinion. It is therefore important to be aware of the implications of the metaphor of war and its oversimplification of complex issues.

## Conclusion

From the perspective of cognitive linguistics, this research investigated the metaphorical use of WAR terms for the Covid-19 pandemic. The detailed analysis of editorials of three renowned Pakistani English Newspapers concluded that WAR terms were widely used in the language of these Newspapers. The study highlighted the explicit deployment of military concepts like BATTLE, ENEMY, WAR, SOLDIERS, FIGHT, and VICTORY to create the conception of WAR against the Covid-19 disease. Furthermore, by combining all these metaphors, the Pakistani Newspapers created a SELF Vs OTHER distinction between the Pakistani people and the medical illness of Covid-19. The inquiry demonstrated that in order to create a sense of urgency among the Pakistani public and to mobilize them against the deadly virus, the military concepts were deliberately engaged to present Covid-19 as an 'alien', 'outsider', as well as an 'enemy' entity.

This research has some implications for future reporting of Covid-19 in Newspaper discourse. Although, in short-term goals, WAR metaphors could create urgency among the public and could mobilize them against the virus; but, in long-term, they breed fear among the audience and cause damage to civil liberty as suggested by Rohela et al. [68] and Panzeri [69] in their studies. For a peaceful and constructive society, considering war as the primary reference point could militarize the community. Therefore, for supporting human rights and building peaceful society in the middle of crisis, it is pertinent to demilitarize our language. Calling

Covid-19 as GLOBAL CRISIS and HEALTH EMERGENCY would be more effective as it would involve HUMANITARIAN metaphors. The study validates the results of Rohela et al. [68] to utilize gentler metaphors maybe from sports and ecology which emphasize human connection and support. In WAR, we need to fight each other; but in EMERGENCY we need to help each other. As, WAR metaphors are hostile and counterproductive; therefore, we need HUMANITARIAN metaphors that could help us in forging an empathetic, compassionate, and peaceful society during pandemic.

Overall, the findings of this study can have several implications for universities, government, and the general public. It can provide insight into how metaphors are used to discuss the global pandemic of Covid-19, and the role of media in this discussion. It can also be used to analyze the potential implications of the metaphors used, and how they can shape public opinion. Furthermore, the findings of this study can also be used to inform public health campaigns and initiatives, as well as policies to help prevent the spread of diseases. Finally, the findings can also be used to educate the public about the importance of using accurate and appropriate language to discuss the pandemic or future threats.

## Supporting information

**S1 Table. Size of PakNCovid-19 corpus.**
(PDF)

**S2 Table. The steps involved in the Pragglejaz Method for Finding Metaphorically Used Words.**
(PDF)

**S3 Table. War metaphors of Covid-19 discourse in Pakistan.**
(PDF)

**S1 Fig. WAR against Covid-19.**
(TIF)

**S2 Fig. Covid-19 as an ENEMY.**
(TIF)

**S3 Fig. Battle against Covid-19.**
(TIF)

**S4 Fig. Doctors as SOLDIERS.**
(TIF)

**S5 Fig. FIGHT against Covid-19.**
(TIF)

**S6 Fig. VICTORY over Covid-19.**
(TIF)

**S1 Data.**
(TXT)

## Author Contributions

**Conceptualization:** Arooj Rana, Tahir Ayoub, Shazia Akbar Ghilzai, Wasima Shehzad.

**Data curation:** Tahir Ayoub, Wasima Shehzad.

**Formal analysis:** Arooj Rana, Shazia Akbar Ghilzai, Wasima Shehzad.

**Investigation:** Arooj Rana, Tahir Ayoub, Shazia Akbar Ghilzai.

**Methodology:** Tahir Ayoub, Shazia Akbar Ghilzai.

**Software:** Arooj Rana, Tahir Ayoub.

**Visualization:** Tahir Ayoub.

**Writing – original draft:** Arooj Rana, Tahir Ayoub.

**Writing – review & editing:** Shazia Akbar Ghilzai, Wasima Shehzad.

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
