## [Decision Letter · Decision Letter 0]

11 Jan 2023

PONE-D-22-30426‘WE ARE AT A WAR AGAINST COVID-19’: A CORPUS-BASED CRITICAL METAPHOR ANALYSIS OF PAKISTANI NEWSPAPERSPLOS ONE

Dear Dr. Ghilzai,

Thank you for submitting your manuscript to PLOS ONE. After careful consideration, we feel that it has merit but does not fully meet PLOS ONE’s publication criteria as it currently stands. Therefore, we invite you to submit a revised version of the manuscript that addresses the points raised during the review process.

The article is interesting in terms of its content, but requires some minor revisions, as suggested by the reviewers. 

In addition to the reviewers' requests, I would invite the authors to add a table (or more) containing quantitative data, as they refer to the use of a qualitative-quantitative method of analysis in the methodology section. Furthermore, I would suggest that the authors update the bibliography, adding in the literature review section not only references to theories on the interpretation of metaphorical language, but also references to other research (and they are numerous) in which other researchers have analysed war metaphor in relation to COVID-19. At the same time, I would suggest inserting in the conclusions (or separately) a section devoted to the discussion of the results also in the light of similar analyses conducted by other scholars.

We look forward to receiving your revised manuscript.

Kind regards,

Ramona Bongelli, Ph.D.

Academic Editor

PLOS ONE

Journal Requirements:

4. Please remove your figures from within your manuscript file, leaving only the individual TIFF/EPS image files, uploaded separately. These will be automatically included in the reviewers’ PDF.

Reviewers' comments:

Reviewer's Responses to Questions

**Comments to the Author**

1. Is the manuscript technically sound, and do the data support the conclusions?

Reviewer #1: Yes

Reviewer #2: Yes

2. Has the statistical analysis been performed appropriately and rigorously? 

Reviewer #1: Yes

Reviewer #2: Yes

3. Have the authors made all data underlying the findings in their manuscript fully available?

Reviewer #1: Yes

Reviewer #2: Yes

4. Is the manuscript presented in an intelligible fashion and written in standard English?

Reviewer #1: Yes

Reviewer #2: Yes

5. Review Comments to the Author

Reviewer #1: This is well-written research. I can follow the process of data collection and analysis with ease. The research background has also been described clearly. The results are presented and discussed rigorously. I only have one recommendation though, even you have stated your research significancy in the conclusion section, if you can suggest practical benefit of your findings to related parties, such as universities, government, that would be better.

Reviewer #2: The article presents new data however the structure of the article does not follow the correct style of writing and many references are archaic.

It will be valid if the findings include statistical values of the issues 'war', (provide the numbers of percentage) of the word existence in the research.

6. PLOS authors have the option to publish the peer review history of their article (what does this mean?). If published, this will include your full peer review and any attached files.

Reviewer #1: No

Reviewer #2: No

---

## [Author Response · Author response to Decision Letter 0]

16 Jul 2023

response to reviewer #1: Thank you for your positive feedback! We appreciate your suggestion to further strengthen the conclusion section. We agree that discussing the practical benefits of our findings to related parties, such as universities and government, would be beneficial. We added this in the conclusion section of our revised paper.

response to reviewer#2: Thank you for your feedback. We understand that the structure of the article does not follow the correct style of writing and that the references are outdated. We have taken your feedback into consideration and have updated the structure of the article to adhere to the correct style of writing. We have also updated the references to more recent sources. Additionally, we have included statistical values of the issues 'war', providing the numbers of percentage of the word existence in the research. We hope that these changes will make the article more comprehensive.

---

## [Decision Letter · Decision Letter 1]

6 Sep 2023

PONE-D-22-30426R1‘WE ARE AT A WAR AGAINST COVID-19’: A CORPUS-BASED CRITICAL METAPHOR ANALYSIS OF PAKISTANI NEWSPAPERSPLOS ONE

Dear Dr. Ghilzai,

Thank you for submitting your manuscript to PLOS ONE. After careful consideration, we feel that it has merit but does not fully meet PLOS ONE’s publication criteria as it currently stands. Therefore, we invite you to submit a revised version of the manuscript that addresses the points raised during the review process.

Dear authors,

the article needs further revision. Please follow the recommendations of the reviewers. In particular, please pay attention to their suggestions regarding the bibliography to be updated. 

We look forward to receiving your revised manuscript.

Kind regards,

Ramona Bongelli, Ph.D.

Academic Editor

PLOS ONE

Reviewers' comments:

Reviewer's Responses to Questions

**Comments to the Author**

1. If the authors have adequately addressed your comments raised in a previous round of review and you feel that this manuscript is now acceptable for publication, you may indicate that here to bypass the “Comments to the Author” section, enter your conflict of interest statement in the “Confidential to Editor” section, and submit your "Accept" recommendation.

Reviewer #3: All comments have been addressed

Reviewer #4: All comments have been addressed

2. Is the manuscript technically sound, and do the data support the conclusions?

Reviewer #3: Partly

Reviewer #4: Partly

3. Has the statistical analysis been performed appropriately and rigorously? 

Reviewer #3: N/A

Reviewer #4: Yes

4. Have the authors made all data underlying the findings in their manuscript fully available?

Reviewer #3: Yes

Reviewer #4: Yes

5. Is the manuscript presented in an intelligible fashion and written in standard English?

Reviewer #3: Yes

Reviewer #4: Yes

6. Review Comments to the Author

Reviewer #3: The paper examine metaphors of COVID-19 in Pakistani newspapers using CMA. The paper is good but requires some modifications before it can be published in a Q1 journal. Below are my main concerns:

1) the authors state: "The main goal is to pursue the way through which the discussion of the

metaphor in language has shifted from its use as an ornamental figure, as a decorative idiom, to an

indispensable cognitive-linguistic tool that connects our cognitive and semantic domains". This has been established a long time ago! the authors need to provide a more convincing argument.

2) Conceptual Metaphor Theory (CMT) is not adopted in its original form anymore; it has been revised and refined even by the original authors themselves. What seems obvious to me is that the researchers are not keeping up-to-date with new advancements in Cognitive Linguistics, see the sources below:

Kövecses, Z. (2016). Conceptual metaphor theory. In Semino, E. & Demjén Z. (eds.), Routledge Handbook of Metaphor (pp. 31-45). NY: Routledge.

Kövecses, Z. (2020). An extended view of conceptual metaphor theory. Review of Cognitive Linguistics, 18(1), 112-130.

Kövecses, Zoltán. (2021). Standard and Extended Conceptual Theory Revisited: Some Definitional and Taxonomic Issues. In Wen, Xu and John R. Taylor (eds.) The Routledge Handbook of Cognitive Linguistics (pp. 191-204). New York: Routledge.

3) The authors are just summarizing previous works on CMA without engaging in any argumentative discussion; I mean why choose CMA? what does this approach aim to reveal? I am afraid that the authors are narrating information without engaging with it, see for example:

Alnajjar, A., & Altakhaineh, A. R. M. (2023). A critical analysis of metaphors used in Arabic and English cosmetics advertisements. Kervan. International Journal of African and Asian Studies, 27(1).

Zibin, A. (2022). The type and function of metaphors in Jordanian economic discourse: A critical metaphor analysis approach. Language Sciences, 93, 101488.

4) I am astonished that through their search on papers that tackled metaphors of COVID-19 in the related literature, the researchers did not come across the following very recent studies; integrating studies that addressed the same topic but in different cultures and languages enables the authors to draw-cross linguistic implications and contribute to the filed of metaphor studies:

Rajandran, K. (2020). A Long Battle Ahead’: Malaysian and Singaporean prime ministers employ war metaphors for COVID-19. GEMA Online Journal of Language Studies, 20(3), 261-267.

Castro Seixas, E. (2021). War metaphors in political communication on COVID-19. Frontiers in sociology, 5, 583680.

Garzone, G. E. (2021). Rethinking metaphors in COVID-19 communication. Lingue e Linguaggi, (44), 159-181.

Zibin, A. (2022). Monomodal and multimodal metaphors in editorial cartoons on the coronavirus by Jordanian cartoonists. Linguistics Vanguard, 8(1), 383-398.

Younes, A. S., & Altakhaineh, A. R. M. (2022). Metaphors and metonymies used in memes to depict COVID-19 in Jordanian social media websites. Ampersand, 9, 100087.

The authors need to cite these papers and engage in a discussion about the similarities and differences between metaphors of COVID-19 in different languages.

5) The section on 'Traditional approaches to metaphor" should be deleted. It has long been established that metaphors are not just figures of speech! that ship has sailed along time ago! the authors should try to keep up-to-date and to provide theoretical implications!

6) In the methodology section, the authors do not explain how they selected the articles? which keywords did they use? the choice of software is also not explained! the authors are obviously following a corpus-based approach but this makes no sense, since the authors did not conduct a pilot study to collect metaphor candidates and then test them against the corpus! see the following:

Zibin, A. (2021). Blood metaphors and metonymies in Jordanian Arabic and English. Review of Cognitive Linguistics, 19(1), 26-50.

I suggest to rectify this problem, now that it is too late to do so, is for the authors to state that they followed a corpus-driven approach (see the reference above for more info). This means that entire corpus was analyzed manually then the software was used to test the frequency of the metaphors.

7) How did the authors extract the conceptual metaphors from the metaphorical expressions?! see the following:

Steen, G. (2007). Finding metaphor in discourse: Pragglejaz and beyond. Cultura, Lenguaje y Representación/Culture, Language and Representation, 5: 9-25.

The authors did not also explain how they avoided data analysis bias?!

8) In the data analysis section, the authors need to provide the examples clearly. For example, Figure 1 is non-readable! where are the metaphors?

9) The discussion section needs to be enriched with relevant studies and a more in-depth discussion of the implication should be provided, see the references provided above.

10) I am surprised that the author conducted no inferential statistical analysis to determine which sub-domains under the war source domain are used the most. This type of analysis can enhance the quantitative analysis done in this paper.

11) The paper should be proofread for language errors.

Reviewer #4: Before submitting my review, I would like to apologize and acknowledge that due to certain problems, my review was completed hastily.

This is a good paper with a very interesting topic, but it can be improved by employing the following comments.

The structure of the paper should be as follows: abstract, introduction, CMT and medical discourse, method and material, data analysis, conclusion.

The literature of the paper should be related to metaphor in medical discourses, and other papers related to politics and etc., should be removed.

Why do the authors focus only on the metaphors that are based on the concept of war? why do they ignore other important metaphors of corona discourse in Pakistan?

They should justify these points in their paper.

I think the authors can also explain other metaphors in the medical discourse of Corona in Pakistan.

The authors say that we study war metaphor, but they should also explain that what is the target domain of war as a source domain?

The concepts such as victory, soldier, enemy and so on are only elaborations of a general conceptual metaphor like controlling corona is a war.

However, when they present these source domains, they should also mention the target domains of them.

They mention the source domains (such as victory) but they don’t discuss the target domains.

In the paper, they should present a finalized table which includes all the source domains and target domains.

The authors can also use MIP to recognize the metaphorical expressions in their corpus.

The authors should clearly write metaphor formula for all the source domains like X IS Y: Corona is enemy

Medical treatment is war

Doctor is a soldier

I think the authors could also explain all the conceptual metaphors of corona in this corpus, but I leave it to them to decide about it or to justify their decision.

7. PLOS authors have the option to publish the peer review history of their article (what does this mean?). If published, this will include your full peer review and any attached files.

Reviewer #3: No

Reviewer #4: No

---

## [Author Response · Author response to Decision Letter 1]

21 Oct 2023

PLOS One

San Francisco, California, US

Dear editors,

We thank the reviewers for their generous comments on the manuscript and have edited the manuscript to address their concerns. 

In particular, we have added convincing arguments related to recent developments in metaphors and extended conceptual metaphor theory (Kövecses, Z. 2016; Kövecses, Z. 2020; Kövecses, Zoltán. 2021). We have also added recent studies on Covid-19 conceptual metaphors in the study and have drawn cross-linguistic and cultural implications. The section on 'Traditional approaches to metaphor" has been deleted and literature review has been revised by adding more convincing and recent references of studies as per reviewers’ suggestions. Corpus-driven approach has been mentioned in the study and metaphorical expressions were identified using the well-established Pragglejaz Method for Finding Metaphorically Used Words (MIP) (the reference table has been added in Appendix 1). The discussion section is added with relevant studies and a more in-depth discussion of the implication are provided. The structure of the paper is redesigned as follows: abstract, introduction, CMT and medical discourse, method and material, data analysis, conclusion. The literature of the paper is rewritten by adding metaphor in medical discourses, and other papers related to politics and etc., have been removed. Moreover, justification for choosing war metaphors have also been added under a separate heading. Metaphor formula for all the source domains have also been added in a tabular form. 

We believe that the manuscript is now suitable for publication in PLOS One. Regards , Arooj Rana (On behalf of all authors.)

---

## [Editor Report · Decision Letter 2]

27 Oct 2023

PONE-D-22-30426R2Metaphorical Framing of the COVID-19 Pandemic in Pakistan: A Corpus Driven Critical Analysis of War Metaphors in News MediaPLOS ONE

Dear Dr. Ghilzai,

Thank you for submitting your manuscript to PLOS ONE. After careful consideration, we feel that it has merit but does not fully meet PLOS ONE’s publication criteria as it currently stands. Therefore, we invite you to submit a revised version of the manuscript that addresses the points raised during the review process.

The article is undoubtedly much improved as a result of the additions made. 

I would still ask you to fix a few small things:Carefully review the cited bibliography and numbering. For example, missing from the bibliography is Philips 2017, which you cite in the textthe word Table is sometimes used in capitals and sometimes in lower case. Specifically, Table 3 is presented in the text as 03. You write that there is Table 2 in the appendix and instead it is numbered as 1. I would also ask you to move Table 2 in the text so that the reader is facilitated. The methodology section does not state how many researchers analysed the corpus to identify metaphors, nor does it state whether or not an index of agreement was calculated after an individual analysis or whether the researchers instead collectively examined the articles and discussed the identified cases. I would specify this.Finally, I would pay attention to verb tenses. I imagine that the paper was written some time ago (when the pandemic was still going on) and in some sentences it still seems to be. 

We look forward to receiving your revised manuscript.

Kind regards,

Ramona Bongelli, Ph.D.

Academic Editor

PLOS ONE
---

## [Author Response · Author response to Decision Letter 2]

27 Nov 2023

Dear editors,

We thank the Academic Editor for the generous comments on the manuscript and have edited the manuscript to address the concerns. 

In particular, we have edited the numbers of citations and bibliography. Moreover, the Table Numbers have been carefully edited and a similar patter have been followed for citing all the tables. Table 2 have also been moved inside the text for readers’ facilitation. While addressing the third concern of researcher agreement, the method and index of agreement have been added in the research paper. Verb tenses have also been edited as per the editor’s demands. 

We believe that the manuscript is now suitable for publication in PLOS One. 

Regards,

Arooj Rana,

---

## [Editor Report · Decision Letter 3]

4 Dec 2023

PONE-D-22-30426R3Metaphorical Framing of the COVID-19 Pandemic in Pakistan: A Corpus Driven Critical Analysis of War Metaphors in News MediaPLOS ONE

Dear Dr. Ghilzai,

Thank you for submitting your manuscript to PLOS ONE. After careful consideration, we feel that it has merit but does not fully meet PLOS ONE’s publication criteria as it currently stands. Therefore, we invite you to submit a revised version of the manuscript that addresses the points raised during the review process.

Dear authors,

thank you for making the suggested changes. 

With respect to the concordance index, there is no need to explain how it is calculated, because those who use it know how it is done. 

In my previous comments, I asked whether it was calculated or not, but there is no need to include it if it has not been calculated and the agreement was reached by discussing the doubtful cases collectively.

K is calculated immediately after the individual analysis, and not after collaborative discussion (as it sounds from your description) to see if the grid used is shared or if it necessary to modify it. You do not calculate K after the doubtful cases have been discussed because obviously at that point the agreement index will be equal to 1 (i.e. 100%).

I would therefore ask you to fix this section by simply saying that after an initial individual analysis, you discussed the doubtful cases and came to an agreement. That is also perfectly fine. What I asked for was to specify how the agreement was reached.

We look forward to receiving your revised manuscript.

Kind regards,

Ramona Bongelli, Ph.D.

Academic Editor

PLOS ONE
---

## [Author Response · Author response to Decision Letter 3]

17 Dec 2023

Dear Reviewers, 

The concordance Index section is modified as suggested on Dec. 4, 2023. We have that the manuscript is now suitable for publication in PLOS ONE. Thanks for your cooperation. 

best regards, Shazia

---

## [Editor Report · Decision Letter 4]

29 Dec 2023

Metaphorical Framing of the COVID-19 Pandemic in Pakistan: A Corpus Driven Critical Analysis of War Metaphors in News Media

PONE-D-22-30426R4

Dear Dr. Ghilzai,

We’re pleased to inform you that your manuscript has been judged scientifically suitable for publication and will be formally accepted for publication once it meets all outstanding technical requirements.

Kind regards,

Ramona Bongelli, Ph.D.

Academic Editor

PLOS ONE
---

## [Editor Report · Acceptance letter]

8 Aug 2024

PONE-D-22-30426R4 

PLOS ONE

Dear Dr. Ghilzai, 

I'm pleased to inform you that your manuscript has been deemed suitable for publication in PLOS ONE. Congratulations! Your manuscript is now being handed over to our production team.

Kind regards, 

on behalf of

Professor Ramona Bongelli 

Academic Editor

PLOS ONE